# Research on Employee Innovation Ability in Human–Machine Collaborative Work Scenarios—Based on the Grounded Theory Construct of Chinese Innovative Enterprises

**DOI:** 10.3390/bs15070836

**Published:** 2025-06-20

**Authors:** Baorong Guo, Xiaoning Liu, Shuai Liao, Jiayi Hu

**Affiliations:** 1School of Business, Guilin University of Technology, Guilin 541004, China; br_g@glut.edu.cn (B.G.); xiaoningliu2003@163.com (X.L.); 2020046@glut.edu.cn (J.H.); 2Guangxi Research Think Tank on Science and Technology Innovation of Resources and Environment, and Green Low-Carbon Development, Guilin 541004, China

**Keywords:** digital economy, human–machine collaboration, grounded theory

## Abstract

Against the backdrop of the booming digital economy, innovation has emerged as the core driving force for enterprise development, with employees’ innovative capabilities serving as a key competitive advantage for innovative enterprises. Adopting grounded theory as the methodological framework, we obtain multi-source data to investigate the factors influencing employees’ innovative capabilities and their underlying mechanisms. Furthermore, we develop a theoretical model elucidating the formation mechanism of employees’ innovative capabilities in human–machine collaboration contexts, identifying four core dimensions—innovation drivers, human–AI collaboration patterns, knowledge conversion pathways, and technological breakthroughs—that dominantly shape these capabilities. Thus, we reveal that the formation of innovative capabilities constitutes a dynamic interplay of technology empowerment, cognitive restructuring, and collaborative reinforcement and demonstrate its spiral progression characterized by “triggering, collaboration, and iteration”. This research not only contributes to academic discourse but also offers actionable theoretical and practical insights for innovative enterprises to enhance employees’ innovative capabilities, thereby fostering sustainable development in global competition.

## 1. Introduction

In the era of digital transformation and intelligence-driven innovation, organizations are increasingly exploring human–machine collaboration as a strategic approach to augment employees’ innovation capabilities. The extant literature has established that human capital constitutes the core asset for technological advancement and sustained competitive advantage in innovative enterprises, with employees’ creative capacity serving as a critical determinant of organizational sustainability ([4]). Innovation capability is defined as an organization’s capacity to generate, implement, and sustain novel solutions that create value, encompassing technological, process, and organizational innovations. The pervasive integration of artificial intelligence (AI) across industries has transformed traditional human-led workflows into symbiotic human–AI systems characterized by complementary capabilities and synergistic efficiency ([11]). This paradigm shift not only alleviates employees from repetitive data-intensive tasks—thereby liberating cognitive resources for complex decision-making and creative problem-solving—but also introduces transformative challenges to cognitive architectures, collaborative mechanisms, and institutional frameworks ([25]). Digital transformation has reshaped the dynamics of organizational innovation, with human–machine collaboration emerging as a critical yet underexplored frontier. Although artificial intelligence reduces cognitive workload through automation, its impact on the formation of employees’ dynamic innovation capabilities remains fragmented. Existing studies either focus on static technology adoption or isolated behavioral factors, neglecting the interactive mechanisms between AI-driven tools and human cognitive restructuring.

Within China’s distinctive digital ecosystem and socio-cultural context, governmental and industrial commitments to AI adoption have intensified significantly. Following the 2017 *Next Generation Artificial Intelligence Development Plan*, accelerated development of humanoid robotics and intelligent systems has been observed across manufacturing, healthcare, and education sectors ([27]). The 2023 *Innovation Development Guidelines for Humanoid Robots* issued by China’s Ministry of Industry and Information Technology outlines a technical roadmap emphasizing “integration of large AI models with motion control algorithms”, establishing comprehensive innovation platforms that bridge between high-level cognitive architectures (“big brain”) and low-level operational systems (“little brain”). This dual impetus of policy support and technological advancement presents enterprises with pressing challenges in optimizing human–AI collaboration through institutional innovation, knowledge governance, and process reengineering to fully harness employees’ creative potential.

Emerging empirical evidence suggests that human–AI collaboration significantly reduces cognitive workload, thereby enhancing proactive innovation behaviors. Particularly, AI literacy emerges as a critical moderator—employees with advanced technical competencies tend to perceive AI systems as collaborative partners, facilitating cross-functional coordination and experimental learning ([17]). Grounded in the Conservation of Resources theory, these findings refine our understanding of micro-level mechanisms through which human–machine interaction shapes innovative behaviors, providing crucial theoretical foundations for investigating capability formation in collaborative intelligence contexts.

Despite the growing adoption of AI in workplaces, there remains a notable research gap in understanding how human–machine collaboration specifically shapes employees’ innovative capabilities. Prior studies have largely examined static technology adoption or isolated human factors but have overlooked the interactive mechanisms by which AI tools and human cognition jointly foster innovation.

In summary, against this backdrop, as previously outlined in Table 1, this study aims to construct a theoretical model of employee innovation capability in human–AI collaborative work scenarios using grounded theory based on data from Chinese innovative enterprises, reveal the dynamic mechanism through which AI enhances innovation capability by integrating AMO theory and COR theory to bridge theoretical fragmentation, and provide empirical insights for organizations to design AI–human collaboration strategies that align with China’s contextual characteristics. To address this gap, the present study’s objective is to construct a grounded theory model explaining the formation of employee innovation ability in human–machine collaborative work scenarios. In other words, we ask How do human–AI collaboration processes contribute to the development of employees’ innovation ability? By answering this question, our research seeks to fill the void in the literature and provide both theoretical and practical insights into innovation management in the AI era.

## 2. Literature Review

Scholars have conducted extensive research on how to promote employees’ innovative behaviors from the perspectives of social responsibility, performance incentives, and organizational behavior. In the context of social responsibility and employee innovation, [22] ([22]) dedicated themselves to exploring the perception of corporate environmental responsibility and deeply investigated the internal mechanisms influencing employees’ green innovative behaviors, revealing the correlation path between the fulfillment of corporate environmental responsibility and employees’ green innovation. Green innovative behaviors refer to employees’ environmentally sustainable creative activities, such as developing energy-efficient AI algorithms or optimizing resource allocation processes through human–machine collaboration, as defined by [22] ([22]). [11] ([11]) proposed the incentive effect of employment stabilization policies on employees’ innovative activities and analyzed their mechanisms, providing a theoretical basis for how policies can promote employee innovation. From the perspective of performance incentives, [17] ([17]) explored the impact of employee incentives and organizational performance on employees’ innovative capabilities, constructed a performance evaluation system with employees’ innovative capabilities as the core, and emphasized the critical role of performance incentives in enhancing employees’ innovative capabilities. From the organizational behavior perspective, [10] ([10]) investigated the driving mechanisms behind the interactive effects of multi-factors influencing employees’ green innovative behaviors based on the Ability–Motivation–Opportunity (AMO) theoretical framework, providing a systematic framework for understanding the impact of organizational behavior factors on employee innovation. [12] ([12]) discussed the capability quality of enterprise employees in the creation and implementation of open innovation, focusing on the role of employees’ capability quality in innovation within organizational behavior. [18] ([18]) examined the impact of employee protection on corporate innovation capabilities, cutting into the relationship between organizational behavior and innovation from the perspective of employee rights protection.

As research has deepened, scholars have begun to focus on individual factors, leadership factors, and organizational factors. For example, [9] ([9]) studied the impact of artificial intelligence innovation on the growth of logistics enterprises and the boundary effects of market competition. Although their research primarily focuses on the enterprise level, it provides an indirect reference for understanding the role of individuals in an AI innovation environment. [30] ([30]) systematically investigated the potential impacts and mechanisms of enterprises’ AI technological innovation on supply chain concentration, involving the correlation between individual innovative behaviors in supply chains from the perspective of technological innovation. [6] ([6]) analyzed how AI technologies influence internal salary gaps by reshaping job tasks, providing a contextual basis for studying the relationship between individuals’ work status and innovation motivation in technological innovation environments. [20] ([20]) and [28] ([28]) constructed a sequential curve mediation model based on the theory of emotions as social information to explore the relationship between leaders’ angry expressions and employees’ deviant innovation under different degrees of superior organizational integration. This study deeply revealed the complex impacts of leadership emotional expressions on employees’ innovative behaviors. [9] ([9]) explores how employee-generative AI collaboration affects work-family outcomes via relationship instrumentality theory. This study examines its antecedents and outcomes using the Conservation of Resources Theory, highlighting AI’s role in enhancing proactive behavior. Green innovation herein denotes organizational innovation outcomes with environmental benefits, such as low-carbon technology adoption or circular economy solutions enabled by human–AI collaboration, consistent with the framework proposed by [10] ([10]). By integrating leadership and organizational factors, this research expands the research dimensions of leadership’s influence on employee innovation. Unfortunately, these theories remain relatively independent and lack unified integration. Most of them are static analyses that collect data at a single point in time to study relationships between variables. In reality, with frequent changes in enterprise development and market environments, the lack of systematic theoretical research makes it difficult to provide real-time and effective guidance for enterprises to continuously enhance employees’ innovative capabilities ([21]).

In terms of research methods, the existing literature primarily explores multiple dimensions such as innovation influencing factors, management mechanism construction, and policy effect analysis using various methodologies. For example, scholars like [8] ([8]) employed factor analysis to deeply dissect the transmission path of reciprocal preferences on enterprises’ technological innovation capabilities, revealing the mechanism through which cooperative reciprocal psychology among organizational members influences technological R&D investment, achievement transformation, and other links. [3] ([3]) and others systematically constructed an innovation management mechanism centered on creating a corporate innovation atmosphere and stimulating employees’ innovation willingness through regression analysis. [11] ([11]) and others empirically tested the impact of employment stabilization policies on employees’ innovative behaviors using a benchmark regression model, discovering that policy stability indirectly promotes innovation output by alleviating employees’ career anxiety and enhancing their sense of innovation security. [1] ([1]) and others focused on the correlation between enterprises’ knowledge management capabilities and employees’ innovative behaviors through empirical research, confirming that knowledge sharing, integration, and application capabilities are key factors driving employees’ innovative practices. Additionally, [24] ([24]) explored the influence mechanism of psychological capital and transformational leadership on employees’ innovative behaviors through reliability testing combined with empirical analysis.

In view of this, this study focuses on human–machine collaborative work scenarios in the digital economy, taking innovative enterprises as the research objects. Through qualitative research, it breaks through the static and fragmented limitations of existing studies, systematically analyzes the formation mechanism of employees’ innovation capabilities, constructs a mechanism model for the formation of employees’ innovation capabilities in human–machine collaborative scenarios, and reveals the laws of technology-enabled employee innovation in human–machine collaboration. The aim is to inject a dynamic collaborative perspective into the theory of employee innovative behavior, enrich the theoretical system of innovation management in the AI era, and further assist enterprises in unleashing employees’ innovation potential through human–machine collaboration in the digital economy, improving technological iteration efficiency and global competitiveness and providing a theoretical reference for policymakers to improve the innovation ecosystem in the AI era.

## 3. Research Design and Data Analysis

### 3.1. Research Methods

Grounded theory is employed as the research methodology for this study for three key reasons. First, grounded theory, with its scientific and rigorous methodological system, has accumulated abundant application achievements in multidisciplinary fields, such as management and sociology, at home and abroad, demonstrating significant effectiveness in research on sub-disciplines, like innovation management and organizational behavior. Its academic authority and reliability are widely recognized by the academic community, providing a strong guarantee for the scientific validity and generalizability of research outcomes. Second, given that we focus on the abstract topic of employees’ innovation capabilities in human–machine collaborative work scenarios—an area where a mature theoretical system has not yet been formed—it is necessary to construct an innovative theoretical framework by deeply mining enterprise practice phenomena. The exploratory nature of grounded theory aligns with the reverse research logic of “from practical phenomena to theoretical abstraction”. Meanwhile, its high compatibility with longitudinal case studies can fully leverage the unique advantages of the longitudinal tracking data of two enterprises in this study to reveal dynamic evolution mechanisms. Third, the unique continuous comparative analysis method of grounded theory can systematically sort out the correlations and differences among multi-source data, including enterprise interview records, internal documents, industry reports, and literature research results. This data inclusivity and methodological flexibility not only help us comprehensively capture the complex essence of research phenomena but also enable timely optimization of analysis strategies based on new discoveries during the research process, ensuring the integrity and accuracy of theoretical construction. The grounded theory used in this study mainly includes five operational procedures ([16]), and the research process is shown in Figure 1 ([14]).

### 3.2. Research Process

#### 3.2.1. Data Collection and Sources

To minimize the subjectivity of qualitative data, we collected information from multiple channels for cross-validation, forming an “evidence triangle” ([26]) to enhance the reliability and validity of the research. ① Official media data collection: This primarily included leader interviews, corporate official websites, and mainstream media reports, aiming to comprehensively understand the sample enterprises. ② Academic literature: Theoretical frameworks were built by retrieving Chinese and English core academic studies. ③ Semi-structured interviews: Interview outlines were developed through a comprehensive analysis of existing data, followed by interviews with mid-to-senior managers and domain experts from different functional departments of the sample enterprises. During the interviews, collected materials were validated. Additionally, after the interviews, follow-up communications via phone, email, and QQ were conducted to supplement information, re-verify relevant details, and solicit feedback on the research findings ([2]).

To ensure the comprehensiveness and depth of the data, the study employed a multi-stakeholder interview approach, targeting different layers of personnel to gather diverse perspectives. The interview subjects covered a wide range of categories, each contributing unique insights to the research, as detailed in Table 2.

#### 3.2.2. Data Coding Process

Grounded theory employs a highly systematic procedure for data selection and analysis. When researchers effectively implement these procedures, they can achieve high research standards that satisfy the generalizability, replicability, accuracy, rigor, and verifiability of research findings ([5]). We strictly followed the technical procedures for hierarchical coding. ① Coding team establishment: To avoid subjective biases in coding due to coders’ knowledge structures, the author formed a coding team with a doctoral student specializing in innovation management and a senior corporate manager. Each member coded independently and maintained memos throughout the process ([23]). ② Research notebook completion: Detailed records were kept of discussion contents and revision processes, accumulating research insights according to the principle of “writing down everything” ([15]). ③ Repeated comparative analysis: Comparative analysis was integrated throughout the grounded analysis. When new concepts or categories were identified, they were compared with existing ones, and reclassification was conducted as necessary. ④ Reliability and validity testing: Data from cases and interviews were hierarchically coded, while literature research was primarily used to test the saturation of the constructed theory. It should be noted that for coding issues and disagreements, we achieved consensus through methods such as re-coding with returned data, spiral repeated comparison, and joint discussion. The coding in this study included three steps, ① open coding, ② axial coding ([29]), and ③ selective coding, which aimed to explore the core characteristics and internal logical relationships of human–machine collaborative innovation and construct a multidimensional feature system. Additionally, theoretical saturation testing was performed to determine whether new categories emerged through literature research and to revise the established theoretical model. The specific steps are as follows:**① Coding Team Establishment:** To minimize subjectivity, a three-member team was formed:Member 1:A PhD in Management with 8 years of experience in qualitative research and expertise in grounded theory applications who has led the design of coding protocols and served as the primary coder for 30% of the data (45 interviews).Member 2:A PhD in Management specializing in innovation management with multiple publications on organizational innovation using grounded theory who has independently coded 30% of the data (45 interviews) and cross-validated coding consistency.Member 3:Responsible for data management and preliminary analysis, participated in coding 40% of the data (60 interviews) under supervision with a focus on initial concept extraction, and independently coded 20% of the data (30 interviews), maintaining detailed memos.**② Inter-Coder Reliability Testing:**Procedure: We randomly selected 40% of initial codes (75 out of 187 codes) for dual coding by two independent coders.Metrics: Inter-coder reliability (IAR) was calculated using Scott’s pi coefficient, with a threshold of ≥0.80 considered acceptable.Results: The average Scott’s pi was 0.83, indicating strong agreement. Discrepancies (e.g., 8% of codes) were resolved through three rounds of spiral comparison and joint discussion, with a final consensus reached by all team members.**③ Research Notebook Maintenance:**Detailed records were kept of coding disagreements (e.g., whether to categorize “data gaps” under “innovation drivers” or “tool limitations”), revision processes, and theoretical insights, following [15]’s ([15]) “write everything down” principle.**④ Theoretical Saturation Testing:**Method: Data collection ceased when no new concepts emerged from five consecutive interviews.Saturation Coefficients:Open coding: 0.925 (11 free concepts out of 146 total);Axial coding: 0.921 (3 free concepts out of 38 total);Selective coding: 1.000 (no new core categories);Confirming theoretical saturation (P9).

## 4. Data Analysis

### 4.1. Open Coding

To clearly demonstrate the process of labeling raw data, forming primary codes, and refining codes into concepts, we list the refinement process of partial codes and concepts in Table 3 ([7]). As Table 4 exemplifies, the study adopted a scientific and rigorous approach to gradually abstract raw data into theoretical concepts. This process not only ensures that concepts are grounded in actual data with a solid real-world foundation but also clearly illustrates the logical derivation from specific phenomena to abstract concepts, enhancing the persuasiveness of the research findings.

During the open coding stage, we systematically combed through and repeatedly analyzed the data paragraph by paragraph, extracting initial concepts related to the theme of human–machine collaborative innovation and categorizing them into categories based on semantic features. The coding process emphasized the fidelity of raw data to ensure that the extracted concepts accurately reflected the core content. Our consistency testing strictly followed the guidelines of [13] ([13]). After initial testing and iterative revisions of the codes, a total of 187 codes were formed, with a consistency test result of 83%, meeting the standard for good reliability. We implemented four-layer quality control.

① Line-by-line parsing: Raw data were coded line by line, with 187 initial codes grouped into 12 categories (e.g., “Data Constraints”) via constant comparative analysis. For example, “Data Gaps Inspire Innovation” (C1) emerged from 18 respondent mentions of data scarcity (Table 3).② Literature cross-checking: Axial coding validated core categories (e.g., “Tool Efficiency”) against AI productivity studies ([11]).③ Causal mapping: Selective coding mapped category relationships (e.g., “Innovation Drivers ↔ Collaboration Mode”) with at least three data instances per link.④ Theoretical saturation: Testing showed coefficients of 0.925 (open), 0.921 (axial), and 1.000 (selective), confirming no new concepts ([14]).

On this basis, we further refined the 187 codes, identifying 12 key categories and their subordinate initial concepts while recording their frequencies. These categories cover multiple dimensions, including data, tasks, human–machine roles, and innovation triggers. As shown in Table 5, Categories with higher occurrence frequencies revealed that issues related to data and tools are particularly prominent in actual human–machine collaborative innovation processes, serving as critical factors influencing innovation. The details are as follows.

### 4.2. Axial Coding

Open coding helps us systematically understand the essential characteristics and interrelationships of key factors in human–machine collaborative innovation, but it is insufficient for clustering the categories formed through open coding. In this study, axial coding was used to reorganize these categories in new ways after open coding ([19]), further identifying the internal correlations between categories and constructing a hierarchical theoretical framework. This provides a structured foundation for in-depth exploration of the formation mechanism of employees’ innovation capabilities in human–machine collaborative scenarios. At this stage, through induction and aggregation of the twelve categories, we further identified their internal relationships and integrated them into nine core categories. These core categories are innovation-driving factors, human–machine collaboration models, tools and effectiveness evaluation, knowledge transformation and management, institutional and mechanism design, technological breakthroughs and development, innovative thinking and strategies, conflict resolution and collaboration paradigms, and capability enhancement and challenges—are presented in Table 6, which details the classification of each category.

Axial coding reorganized categories into nine core constructs, with disputes resolved through structured arbitration—for instance, the team debated whether “AI’s rigid output formats” belonged to “Tool Limitations” or “Cognitive Conflicts” during this stage. The resolution process involved reviewing original interview data (e.g., Respondent 2’s comment that “The AI’s standardized reports limited our creative problem-solving”), referencing the grounded theory literature ([15]) to emphasize data-driven coding and reclassifying the code under “Tool Limitations” with a supporting memo validated by 80% of the team. The coding manual was updated three times throughout the process; for example, after analyzing the 20th interview, the category “Expectations for Future Systems” was split into “Interaction Upgrading” and “Autonomous Learning” to better reflect emerging themes.

### 4.3. Selective Coding

To further uncover the internal logic of employees’ innovation capabilities in human–machine collaborative enterprises, selective coding is conducted to clarify the typical relational structures among core categories and interpret their intrinsic logical connections. This clarification of internal logic provides a clear theoretical framework for enterprises and organizations to promote human–machine collaborative innovation practices, enabling the formulation of more targeted strategies and measures. The results of selective coding are presented in Table 7, which details the typical relational structures and their connotations.

### 4.4. Saturation Testing

We ensured the sufficiency of theoretical construction through saturation testing at three coding levels. As presented in Table 8, the results showed that the saturation coefficients for the open coding level, axial coding level, and selective coding level were 0.925, 0.921, and 1.000, respectively, all meeting the theoretical saturation requirements of grounded theory. This indicates that with the progression of coding, fewer new concepts emerged, and the theoretical framework gradually approached saturation. The current coding system can comprehensively cover the key aspects of the research phenomenon, and the contribution of research data to theoretical construction has approached its limit, demonstrating that the theory has good stability and reliability. The high saturation coefficients confirm the depth of data mining and the rigor of the analysis process, providing a solid foundation for follow-up research.

## 5. Mechanism and Result Analysis of Influences

Through the three-level coding of multi-source data, this study constructs a “Model of the Formation Mechanism of Employees’ Innovation Capabilities in Human-Machine Collaborative Work Scenarios”. The explanatory power of these core categories is further validated, as shown in Table 9. The model reveals that in a work environment with deep AI involvement, the formation of employees’ innovation capabilities is a dynamic interactive process comprising technology, empowerment–cognitive, reconstruction–collaboration, reinforcement, and it is jointly influenced by four core categories: innovation-driving factors, human–machine collaboration models, knowledge transformation pathways, and technology breakthrough directions. As Figure 2 indicates, this process exhibits the spiral upward characteristics of “trigger-collaboration-iteration”—factors such as data gaps and technical limitations activate innovation needs (trigger stage). Program iteration is achieved through the human–machine division of labor and knowledge transformation (collaboration stage), and innovation outcomes are fed back into technical tool optimization (feedback stage). Ultimately, this forms a positive cycle of “capability improvement-technology upgrading”, driving the continuous advancement of employees’ innovation capabilities.

Our findings also align with the Ability–Motivation–Opportunity (AMO) theory, which posits that organizational innovation capability is rooted in employees’ combined attributes of ability, motivation, and access to opportunities. Specifically, AI tools enhance employees’ technical capabilities by automating data analysis and routine tasks, such as coding and report generation. As frontline R&D staff noted in the cases, AI “streamlines data processing, allowing us to apply specialized knowledge to complex problems” (Respondent 17), thereby freeing cognitive resources for higher-order innovation.

Drawing from the Conservation of Resources (COR) theory, AI’s role in reducing cognitive burden also aligns with AMO’s motivation dimension. When employees are relieved of tedious workloads, their intrinsic motivation to engage in creative tasks increases. For example, a marketing manager stated “AI handles repetitive customer segmentation, so we can focus on brainstorming innovative campaign strategies” (Respondent 9), reflecting how resource preservation via AI enhances motivational states.

AMO theory emphasizes that organizational structures must provide opportunities for employees to apply their abilities. In our model, AI acts as an “opportunity enabler” by reducing time spent on routine work—such as a 40% reduction in data collation time across cases—thereby creating temporal and mental space for innovation. This aligns with the AMO framework, where AI-driven efficiency directly correlates with increased opportunities for creative problem-solving.

By integrating AMO theory, we clarify that the “technology empowerment” mechanism not only enhances employees’ practical ability to innovate but also sustains their motivational drive and expands their operational scope. This theoretical linkage bridges grounded theory findings with established organizational behavior frameworks, demonstrating that AI’s impact on innovation capability is multidimensional and aligns with classic human resource management theories.

The following elaborates on the core processes and key mechanisms.

### 5.1. Core Processes: Interaction of Three Dimensions

#### 5.1.1. Technology Empowerment Drives the Balance of Innovation Demand

In human–machine collaboration, employees’ satisfaction with technological tools in meeting their needs exhibits characteristics of “supply-demand balance”.

(1)Internal Supply–Demand Balance: This balance focuses on the dynamic matching mechanism between the functional adaptability of technological tools and employees’ innovative task requirements. The complex decision support capabilities of AI tools assist employees in completing high-complexity decisions within a limited time by rapidly analyzing the advantages and disadvantages of multidimensional data options. Their cross-domain data integration functions break down data silos to achieve integrated analysis of dispersed data, providing comprehensive support for innovation. The core of this balancing mechanism lies in the cognitive resource release effect of technological tools—automating low-level tasks to allow employees to reallocate attention to high-order cognitive activities, such as creative ideation and value judgment. The high alignment between technological supply and demand is continuously optimized through a dynamic adjustment mechanism. As task complexity grows exponentially, the system must possess adaptive learning capabilities, such as dynamically adjusting the weight of functional modules via machine learning algorithms to ensure tool efficiency evolves in tandem with task requirements. When the functions of technological tools are highly aligned with employees’ innovative task needs, the internal supply–demand balance is achieved. Once formed, this balance positively impacts employees’ innovation efficiency, enabling them to utilize resources more effectively in the innovation process and propose more creative and valuable solutions. Forming an internal supply–demand balance enhances innovation efficiency.(2)External Supply–Demand Balance: This balance emphasizes the coupling relationship between organizational intelligent systems, employees’ technical literacy, and task complexity. The operational friendliness and interface scalability of intelligent systems directly influence the depth of technology adoption, while differences in task complexity require systems to have differentiated capabilities—simple tasks need basic functional support, while complex tasks rely on multimodal data analysis, real-time collaboration, and precise simulation prediction capabilities. The adaptation between technological supply and demand is dynamically optimized through an organizational learning feedback mechanism. The continuous collection and analysis of technology usage data can identify functional redundancies or capability gaps, thereby driving system iteration and upgrading. Achieving external balance not only enhances employees’ perception of technological empowerment but also promotes the sustainable evolution of human–machine collaboration systems through the closed-loop logic of “demand identification-technical response-efficiency improvement”. Considering task complexity, different innovation tasks impose varying requirements on intelligent systems. When intelligent systems can perfectly align with employees’ technical literacy and task complexity, employees will clearly perceive the adaptability between technological supply and task demands during innovative work using intelligent systems, thereby strengthening their perception of innovation support and stimulating innovation enthusiasm.

#### 5.1.2. Collaboration Enhancement Promotes Knowledge Transformation and Conflict Resolution

In human–machine collaboration, the knowledge flow and collaboration mechanisms directly influence the externalization of innovation capabilities.

(1)Tacit Knowledge Explicitization: Human–machine interaction transforms employees’ implicit knowledge, such as debugging strategies and experiential intuition, into reusable knowledge assets, enabling continuous accumulation in organizational knowledge bases. This process follows a dual-coding mechanism. Human experience is converted into machine-readable structured data through interactive interfaces, while machine-generated feedback expands human cognitive boundaries, fostering a “dual-track thinking” model. Within this framework, employees shift from traditional “executor” roles to “value architects”, focusing on demand definition and ethical judgment—leveraging deep business insights to accurately identify user pain points and assigning clear value anchors to innovation activities through goal-oriented path design.(2)Conflict Resolution and Paradigm Innovation: Conflicts between technical rationality and business logic constitute a driving force for innovative breakthroughs. When AI-recommended solutions contradict business rules or ethical standards, employees must employ strategies, such as multidimensional trial and error and user-centered design to reconcile contradictions. Conflict resolution relies on a multi-criteria decision-making framework. Employees must dynamically filter and iteratively optimize massive generated solutions across dimensions including technical feasibility, economic cost, user experience, and ethical compliance. This process drives role transformation—employees evolve from solution executors to decision arbitrators, reconstructing problem-solving pathways through interdisciplinary knowledge integration.(3)In this process, employees’ roles shift from traditional solution executors to decision-makers. They must screen numerous AI-generated proposals, selecting the most suitable ones based on factors such as actual business needs, cost effectiveness, and user experience. Simultaneously, employees must conduct ethical evaluations to ensure that plan implementation does not trigger ethical issues like user privacy violations or social fairness disruptions. This division of labor significantly expands employees’ thinking, fostering interdisciplinary associations and breakthrough innovations. Meanwhile, interdisciplinary thinking breaks down disciplinary barriers, enabling employees to examine problems from diverse perspectives and generate entirely new ideas and solutions, thus driving the realization of breakthrough innovations.

### 5.2. Key Mechanisms: Dual Logic of Innovation Capability Formation

#### 5.2.1. Complementary Substitution Mechanism: Synergistic Enhancement of Technology and Human Resources

The complementary substitution mechanism achieves the organic unity of technological substitution and human capability enhancement through task division topology theory. Artificial intelligence demonstrates inherent efficiency advantages in modular, highly repetitive tasks, with its substitution effect freeing up employees’ creativity to focus on creative generation for unstructured problems. At the same time, data-driven prediction and domain experience judgment form a complementary synergy, constructing a “data-insight” dual innovation path. Data-driven prediction reveals potential patterns through massive data analysis, providing convergent power for innovation directions; domain experience judgment retains path explorability based on professional insights into business scenarios. The dynamic tension between the two forms a driving force for expanding the innovation space. Synergistic enhancement requires integration into a three-dimensional evaluation system—measuring the breakthrough degree through solution novelty, assessing technical implementation potential through feasibility, and examining user perception and commercial returns through value—to ensure that innovation plans have both scientific rationality and market orientation.

#### 5.2.2. Weakening Inhibition Mechanism: Balancing Technological Risks and Innovation Resistance

The weakening inhibition mechanism is designed to mitigate the restrictive effects of rigid technical standardization and employees’ anxiety about capability reconstruction on innovation activities. Excessive technical standardization may lead to path dependence—for example, fixed algorithm frameworks can limit personalized exploration, necessitating flexible mechanisms to grant parameter adjustment authority and embed flexible innovation spaces within standardized processes. To address anxiety about capability reconstruction, organizations need to design immersive collaborative training systems that reduce psychological barriers through simulated human–machine collaboration scenarios while also building dynamic support systems for capability reconstruction. These systems should include comprehensive designs of technical redundancy, decision-making participation, and error tolerance. For instance, allowing employees to customize modifications to certain functions in intelligent systems not only ensures system stability but also provides a safe boundary for trial-and-error innovation, forming a symbiotic relationship between risk buffering and innovation incentives.

### 5.3. Evolution of the Theoretical Model

The formation of employees’ innovation capabilities is not a linear process but exhibits a spiral upward characteristic of “trigger-collaboration-iteration”.

(1)Trigger Stage: Innovation Drivers Activate Innovation DemandIn human–machine collaborative work contexts, innovation drivers, such as data gaps and technological limitations, act as key variables that disrupt the existing innovation supply–demand balance. The incompleteness of data and the limitations of technology in specific tasks pose challenges to existing work models. These challenges prompt employees to reflect on current work processes and technology applications, cognitively stimulating innovation demand. Driven by the pursuit of business goals and expectations for work efficiency improvement, employees keenly perceive the necessity of innovation, thereby igniting the engine of innovative thinking and laying a demand foundation for subsequent innovation activities.(2)Collaboration Stage: Human–Machine Division of Labor and Knowledge Transformation Drive Innovation Plan IterationThe collaboration stage represents the core of human–machine collaborative innovation, where the division of labor between humans and machines and knowledge transformation play critical roles. Based on their respective advantageous attributes, a refined division of labor is implemented. Artificial intelligence undertakes data-intensive and rule-defined tasks, while employees focus on creative, strategic, and emotional interaction tasks. Simultaneously, the tacit knowledge transformation mechanism is activated, enabling the explicitization of employees’ tacit knowledge accumulated through practice via means such as knowledge coding and sharing. Throughout this process, the complementary substitution mechanism remains integral, with the knowledge and capabilities of humans and machines mutually supplementing and substituting each other. Through continuous interaction and integration, this mechanism drives the continuous iterative optimization of innovation plans, gradually evolving toward more innovative and feasible directions.(3)Feedback Stage: Innovation Achievements Feed Back to Form a Positive CycleInnovation outcomes generated from implementing innovation plans serve as crucial feedback to propel the sustained development of human–machine collaboration. These outcomes—such as new algorithms and optimized processes—are integrated into the optimization process of technological tools. The upgraded technological tools further enhance their enabling role in human–machine collaboration, improving employees’ work efficiency and innovation capabilities. With support from superior technological tools, employees can expand the frontiers of innovation and propose higher-quality innovative ideas, thereby forming a positive cycle of “capability enhancement—technology upgrade”. This cyclical mechanism ensures the dynamic balance and continuous evolution of the human–machine collaborative innovation system, continuously propelling employees’ innovation capabilities to higher levels.

## 6. Discussion

### 6.1. Theoretical Contributions

This study deepens the theoretical understanding of the formation of employees’ innovation capabilities in human–machine collaborative innovation, primarily in the following aspects:(1)Constructing a Systematic Theoretical Framework: Through a rigorous three-level coding process, this study has developed a “Model of the Formation Mechanism of Employees’ Innovation Capabilities in Human-Machine Collaborative Work Scenarios”, systematically integrating four core categories: innovation-driving factors, human–machine collaboration models, knowledge transformation pathways, and technological breakthrough directions. Previous research has mostly focused on fragmented aspects of human–machine collaboration, lacking systematic integration. This study fills this gap by providing a structured and systematized theoretical framework for the field, expanding the boundaries of human–machine collaborative innovation theory.(2)Revealing Dynamic Interaction Mechanisms: Centered on technological empowerment, cognitive reconstruction, and collaboration enhancement, this model elaborates, in detail, on the dynamic interaction process of employees’ innovation capabilities in human–machine collaboration scenarios—from triggering and collaboration to feedback. This corrects the static and one-sided interpretations of the formation mechanism of employees’ innovation capabilities in previous studies, presenting the complex correlations and action pathways among various elements from a dynamic perspective. It deepens the understanding of this mechanism and provides more precise theoretical guidance for follow-up research.(3)Enriching the Theory of Innovation Influencing Factors: The study clarifies the critical role of innovation-driving factors in stimulating employees’ innovation demand, as well as the synergistic impacts of human–machine collaboration models, knowledge transformation pathways, and technological breakthrough directions on the formation of innovation capabilities. Different from previous studies that only focused on a single or a few influencing factors, this research comprehensively combed through multiple factors and their interrelationships, enriching the theoretical connotation of influencing factors of employees’ innovation capabilities.

### 6.2. Practical Contributions

The research findings provide practical guidance for enterprises to enhance employees’ innovation capabilities and optimize human–machine collaborative innovation practices:(1)Optimizing Human–Machine Collaboration Strategies: Enterprises can accurately identify innovation-driving factors and reasonably adjust human–machine collaboration models based on the model constructed in this study. Meanwhile, promoting the transformation of employees’ roles from mere executors to decision-making participants can fully unleash the potential of both humans and machines, thereby enhancing the overall innovation efficiency of the enterprise.(2)Facilitating the Cultivation of Employees’ Innovation Capabilities: Enterprises can leverage the model to clarify the key links in cultivating employees’ innovation capabilities. By creating a suitable innovation environment—such as providing abundant data resources and optimizing technological tools—enterprises can stimulate employees’ innovation demand. Additionally, building knowledge sharing platforms can promote the explicitization and dissemination of employees’ tacit knowledge, strengthening knowledge transformation pathways. Establishing effective incentive mechanisms to encourage employees to actively participate in human–machine collaborative innovation practices can achieve the synchronous improvement of employees’ innovation capabilities and corporate innovation performance.(3)Guiding the Application and Upgrading of Technological Tools: Enterprises can make targeted resource investments in the research and development of technological tools based on the guidance of technological breakthrough directions in the model. For example, enhancing innovations in human–computer interaction technology to improve the convenience and intelligence of interaction and promoting the development of artificial intelligence’s autonomous learning capabilities to better adapt to complex and changing work scenarios can provide more robust technical support for employees’ innovation.

### 6.3. Research Limitations and Future Prospects

#### 6.3.1. Research Limitations

(1)Sample Limitations: The data in this study primarily originate from enterprises in specific industries and regions, with limited coverage in terms of industry scope and geographical breadth, which may affect the generalizability of the research findings across different industries, cultural backgrounds, and economic development levels.(2)Incomplete Exploration of Factors: The research mainly focuses on human–machine collaborative innovation at the organizational level, with an insufficient discussion of the impacts of macro-industrial environments and micro-individual psychological factors on employees’ innovation capabilities.(3)Lack of Dynamic Research: The study lacks, to a certain extent, tracking of the long-term dynamic evolution process of human–machine collaborative innovation and fails to fully consider potential changes in the formation mechanism of employees’ innovation capabilities along with technological advancements and organizational changes over time.

#### 6.3.2. Future Prospects

Although this study uses grounded theory for qualitative analysis to deeply explore the essence of phenomena, it has certain subjectivity. Researchers’ personal cognitive limitations, biases in interview subject selection, and subjective judgments during data analysis may affect research results. Follow-up research can expand the sample selection scope to cover more enterprises across different industries, regions, and scales, as well as organizations with diverse cultural backgrounds. Through multi-case studies, large-scale questionnaire surveys, and other methods, the generalizability of the research model can be further tested and improved.

To enhance model validity, expert validation will be incorporated in two stages. First, a theoretical expert panel comprising five scholars specializing in innovation management and AI technology will evaluate the model’s theoretical consistency. Their feedback will be integrated to refine constructs, such as the “technological breakthrough directions” category. Second, practitioner validation will involve 10 industry experts who will assess the model’s practical applicability. This process will provide insights into scaling the framework across diverse organizational contexts, with their feedback documented in a validation report to be appended to future publications.

Additionally, this study has not conducted an in-depth quantitative analysis of the specific action pathways, relationship strengths, and index factor weights among various factors. Future research will strive to introduce quantitative research methods, such as structural equation modeling, to deeply explore the quantitative relationships among factors, clarify action pathways and strengths, determine index factor weights, and further improve the theoretical model and evaluation system for employees’ innovation capabilities in human–machine collaborative work scenarios. Meanwhile, there are plans to expand the research scope by comparing differences among innovative enterprises in different industries and development stages, enhancing the generalizability and application value of research outcomes.

## 7. Conclusions

Against the backdrop of the deep integration of artificial intelligence into the development of innovative enterprises, this study employs grounded theory to conduct an in-depth analysis of the influencing factors, formation mechanisms, and dynamic evolution processes of employees’ innovation capabilities in innovative enterprises, leading to the following key findings:(1)Core Categories Govern the Formation of Innovation Capabilities: The four core categories—innovation-driving factors, human–machine collaboration models, knowledge transformation pathways, and technological breakthrough directions—significantly influence the formation of employees’ innovation capabilities. Specifically, innovation-driving factors stimulate innovation demand through a progressive triggering mechanism involving data gaps, technological limitations, and task ambiguity. Human–machine collaboration models optimize collaboration via a spiral evolution path of the dynamic division of labor, role metaphor, and strategic adjustment. Knowledge transformation pathways promote knowledge flow by adhering to a three-stage model of tacit knowledge explicitization, systematic sedimentation, and innovation diffusion. Technological breakthrough directions provide technical support through a dual-engine driving model of interaction upgrading and autonomous learning. These core categories exhibit high conceptual coverage, strongly validating their dominant role in shaping the formation of employees’ innovation capabilities.(2)Key Mechanisms Influence Innovation Capability Development: The complementary substitution mechanism enhances synergistic efficiency between technology and human resources. Artificial intelligence undertakes repetitive tasks, while employees focus on creative generation, leveraging mutual strengths to achieve complementarity. The weakening inhibition mechanism balances technological risks and innovation resistance. Through flexible mechanisms, training, and other means, it mitigates negative impacts such as excessive technological standardization and employees’ anxiety about technology, ensuring the smooth conduct of innovation activities.(3)Innovation Capabilities Evolve in a Spiral Upward Manner: The formation of employees’ innovation capabilities is a spiral upward process of “trigger-collaboration-iteration”. In the trigger stage, innovation drivers activate innovation demand. In the collaboration stage, innovation plans are iterated through the human–machine division of labor and knowledge transformation. In the feedback stage, innovation outcomes feed back into the optimization of technological tools, forming a positive cycle of “capability enhancement—technology upgrade” that continuously drives the improvement of employees’ innovation capabilities.

This study breaks through the fragmented and static limitations of existing research by constructing a dynamic theoretical framework for the formation of employees’ innovation capabilities in human–machine collaboration scenarios. Future research will conduct quantitative validation through structural equation modeling (SEM) on 300 Chinese innovative enterprises to test the causal pathway of “innovation drivers → collaboration patterns → innovation capabilities”. Longitudinal case studies will also be introduced to track the dynamic evolution of human–machine collaboration, thereby refining the model’s generalizability. It not only enriches the theoretical system of innovation management in the AI era but also provides practical guidance for enterprises. By precisely matching human–machine needs, optimizing collaboration models, strengthening knowledge management, and establishing feedback mechanisms, enterprises can systematically enhance employees’ innovation capabilities and help achieve sustainable development in global competition.

## Figures and Tables

**Figure 1 behavsci-15-00836-f001:**
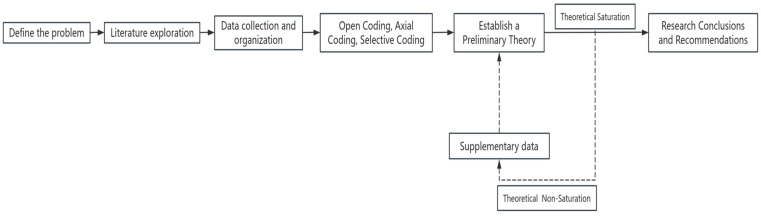
Adapted from [14] ([14]).

**Figure 2 behavsci-15-00836-f002:**
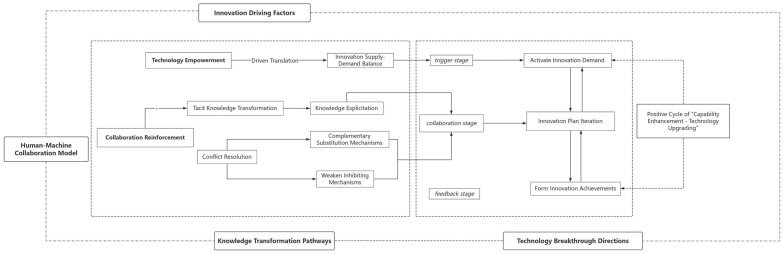
Schematic diagram of the action mechanism.

**Table 1 behavsci-15-00836-t001:** Core differences between this study and previous research.

Dimension	Issues Addressed by Previous Research	New Contributions of This Study
Research Perspective	Focused on single factors (e.g., policy incentives, organizational behavior) affecting employee innovation; static analysis prevailed.	First to construct a dynamic mechanism model in human–machine collaboration scenarios, revealing the “trigger-collaboration-iteration” spiral progression.
Theoretical Framework	Scattered application of theories like AMO theory, lacking systematic integration of human–machine collaboration and innovation capabilities.	Integrated four core frameworks, innovation drivers, collaboration patterns, knowledge transformation pathways, and technological breakthrough directions; proposed the interactive logic of “technology-cognition-collaboration”.
Research Methodology	Predominantly quantitative analysis or single-case studies, lacking qualitative exploration with multi-source data.	Employed grounded theory and multi-source data triangulation (interviews, corporate documents, industry reports); refined dynamic mechanisms through three-level coding.
Practical Guidance	Recommendations focused on fragmented measures.	Provided a four-step framework, “needs matching-collaboration design-knowledge management-technology iteration”, emphasizing organizational learning feedback mechanisms.

**Table 2 behavsci-15-00836-t002:** Data sources.

Number	Data Source	Name Platform	Introduction	Quantity
1	Questionnaire Collection	Semi-structured Interviews for Innovative Enterprises	Cover five fields: e-commerce, logistics, new energy, biomedicine, and retail.	30 questionnaires collected Documents sorted, totaling over 50,000 words
2	News Reports from Chinese Media	Phoenix News Search Engine	Phoenix New Media is a world-leading cross-platform online new media company.	Documents sorted, totaling about 10,000 words
3	Interview Videos of Entrepreneurs	Bilibili Video Search	A well-known video platform and cultural community in China.	Documents sorted, totaling about 60,000 words
4	CNKI (China National Knowledge Infrastructure)	China Academic Journal Network Publishing Database	The world’s leading digital publishing platform.	500 related documents/studies
5	Web of Science	Internationally Renowned Academic Literature Databases	High-quality, multidisciplinary academic journal literature on a global scale.	300 related documents/studies

**Table 3 behavsci-15-00836-t003:** Interview subjects.

Categories of Interview Subjects	Specifically Covered Personnel	The Focus of the Information
Managers	Managers of various business departments, directors, senior corporate leaders	Elucidate Views on Employee Performance and Evaluation Systems from Dimensions of Strategic Planning, Team Management, and Business Advancement
Frontline Employees	Employees in basic business positions such as R&D, technical support, marketing, operations, and administration	Share Intuitive Feelings about the Changes Artificial Intelligence Brings to Work during Direct Participation in Daily Operations
Staff of Human Resources Department	Specialists and heads of modules such as recruitment, training, and performance	Introduce Each Link of the Company’s Existing Employee Evaluation System, as well as the Problems Faced by the System in the Era of Artificial Intelligence and Its Improvement Directions
Experts and Scholars	Experts and scholars in fields such as artificial intelligence technology, organizational behavior, and innovation management	Provide Comprehensive and In-Depth Professional Insights by Virtue of Professional Attainments and Experience to Facilitate Research Advancement

**Table 4 behavsci-15-00836-t004:** Examples of the formation process from codes to concepts (sorted by original data).

Original Data Segment (Code Number and Paragraph Number)	Initial Code	Code	Concept
“I find that data gaps often inspire my creativity. Take the recommendation system as an example. In the past, relying solely on limited user behavior data was like a skilled cook without ingredients—even the most capable person struggles to create something without the right resources” (a1) (Respondent 1, Q2).	AA1 (a1) Innovation Inspired by Data Scarcity	A1 Data-Driven Innovation Trigger	C1 Data Gaps Inspire Innovation
“At first, the software pushed exercises and explanations to children in a fixed course sequence, without considering their actual mastery level at all. I thought we needed to make the software analyze children’s learning data and identify their weak points” (a55) (Respondent 5, Q1).	AA23 (a55) Rigid Tasks Prompt Creative Solutions	A18 Task Personalized Redesign	C6 Unclear Task Objectives
“I think an intelligent system is like a ‘horse’ with strong running ability, capable of quickly handling a large amount of basic calculations. I am the ‘rider’ who needs to control it and guide it in the right direction” (a144) (Respondent 3, Q3).	AA49 (a144) Human Guidance, Machine Execution	A35 Human–Machine Synergy Enhancement	C16 Humans as Decision-Makers
“Our company has implemented an innovation project claim system. As long as you have good ideas for human-machine collaborative innovation, you can claim a project, and the company will allocate personnel and resources to you” (a165) (Respondent 1, Q5).	AA102 (a165) Institutional Support for Innovation	A42 Project-Based Innovation Incentives	C18 Innovation Incentive Mechanisms
“I organized the reasons behind my intuitions, such as overly complex code logic or excessive function calls, into checklists. Then, together with developers, we turned the contents of the checklists into automated test scripts” (a196) (Respondent 1, Q6).	AA221 (a196) Encoding of Tacit Knowledge	A109 Transformation of Experience into System Rules	C47 Encoding of Experience
“The machine’s recommendation focused on a single platform for product promotion. I believed that multi-platform promotion could reach more users. So, we separately collected data on user activity and conversion rates of different platforms” (a376) (Respondent 2, Q7).	AA161 (a376) Resolution of Human–Machine Conflicts	A89 Data-Driven Conflict Resolution	C23 Data-Driven Verification
“In the past, I used to think in a linear way. When I got a task, I would break it down step by step using old methods. But now, after long-term collaboration with machines, my way of thinking has changed significantly” (a392) (Respondent 1, Q8).	AA96 (a392) Transformation of Problem-Solving Approaches	A54 Multidimensional Thinking	C27 Transition to Multidimensional Thinking C32 Flexible Parameter Adjustment
“The company allows us to make custom modifications to 20% of the test cases in the intelligent testing system” (a417) (Respondent 2, Q9).	AA75 (a417) Flexibility within Standardized Systems	A48 Customizable System Functions	

**Table 5 behavsci-15-00836-t005:** Results of open coding analysis.

Number	Category (Frequency)	Initial Concept (Frequency)
1	Data Constraints (48)	Innovation Triggered by Data Gaps (18), Limited Data Availability (15), Need for New Data Sources (10), Challenges in Data Integration (5)
2	Task Ambiguity (36)	Unclear Task Objectives (12), Self-Defined Objectives Required (10), Exploratory Task Design (8), Flexible Task Boundaries (6)
3	Human–Machine Role Division (62)	Humans as Decision-Makers (20), Machines as Auxiliary Tools (18), Dynamic Role Adjustment (14), Synergistic Enhancement (10)
4	Innovation-Triggering Factors (55)	Humans as Decision-Makers (20), Machines as Auxiliary Tools (18), Dynamic Role Adjustment (14), Synergistic Enhancement (10)
5	Tool Efficiency (70)	Information Integration (25), Solution Verification (20), Time-Saving Automation (15), Enhanced Data Analysis (10)
6	Tool Limitations (45)	Rigid Formats (15), Lack of Creative Flexibility (12), Over-Reliance on Standard Outputs (10), Limited Contextual Understanding (8)
7	Institutional Support (60)	Innovation Incentive Mechanisms (18), Cross-Departmental Collaboration (15), Error-Tolerant Policies (12), Knowledge Sharing Platforms (10), Allocation of Innovation Resources (5)
8	Transformation of Tacit Knowledge (52)	Encoding of Experience (20), Transformation of Intuition into Rules (15), Creation of Checklists (10), Automation of Tacit Knowledge (7)
9	Resolution of Cognitive Conflicts (40)	Data-Driven Verification (15), Cross-Comparison of Alternative Solutions (10), User-Centered Compromise (8), Iterative Testing (7)
10	Shift in Thinking Patterns (65)	Transition to Multidimensional Thinking (22), Creativity Inspired by Data (18), Divergent Problem-Solving (15), Enhanced Inspiration Generation (10)
11	Balance between Standardization and Innovation (50)	Flexible Parameter Adjustment (18), Customizable System Functions (15), Creative Autonomy within Standards (12), Adaptive System Design (5)
12	Expectations for Future Systems (58)	Enhanced Interaction Paradigms (20), Understanding of Ambiguous Requirements (15), Brain–Computer Interfaces (12), Multi-User Collaboration (11)

**Table 6 behavsci-15-00836-t006:** Axial coding.

Core Category	Main Category	Category
C1 Innovation-Driving Factors	B1 Data Gap	A2 Situational Stimulation of Innovation Willingness, A12 Innovation-Triggering Conditions, A42 Technological Breakthrough Paths, A62 Multi-source Data Fusion System
	B2 Technical Limitations	A12 Innovation-Triggering Conditions, A32 Innovation-Driving Forces, A42 Technological Breakthrough Paths, A57 Incentives for Innovative Technologies
	B3 Task Ambiguity	A12 Innovation-Triggering Conditions, A22 Problem-Solving Paths, A58 Characteristics of Innovation Tasks
C2 Human–Machine Collaboration Mode	B4 Dynamic Division of Labor	A3 Role Division in Innovation Activities, A13 Human–Machine Role Positioning, A23 Human–Machine Collaboration Mode, A33 Perception–Cognition Division of Labor, A43 Human–Machine Translation Collaboration, A59 Human–Machine Collaboration Division of Labor
	B5 Role Metaphor	A13 Human–Machine Role Positioning, A23 Human–Machine Collaboration Mode, A60 Human–Machine Responsibility Positioning, A63 Dynamic Regulation and Balance System of Human–Machine Collaboration
C3 Tool and Effectiveness Evaluation	B6 Efficiency Improvement	A4 Impact of Tools on Innovation Efficiency, A14 Effectiveness of Collaborative Tools, A44 Effectiveness of Office Tools, A64 Double-Edged Sword Effect of Tools
	B7 Innovation Constraints	A4 Impact of Tools on Innovation Efficiency, A24 Limitations of Testing Tools, A44 Effectiveness of Office Tools, A64 Double-Edged Sword Effect of Tools
C4 Knowledge Transformation and Management	B8 Explicitization of Tacit Knowledge	A6 Transformation of Tacit Knowledge into Innovation Inputs, A16 Transformation of Tacit Knowledge, A66 Systematic Sedimentation of Knowledge Assets
	B9 Encoding of Experience	A16 Transformation of Tacit Knowledge, A36 Optimization of Deployment Processes, A46 Inputs for Process Optimization
	B10 Innovation Diffusion	A45 Innovation Knowledge Management
C5 System and Mechanism Design	B11 Material Incentives	A5 Institutional Design to Promote Innovation, A15 Institutional Guarantee for Innovation, A35 Construction of Innovation Ecosystem, A65 Institutionalized Innovation Ecosystem
	B12 Flexible Mechanisms	A9 Reconciliation between Standards and Innovation, A19 Reconciliation between Standardization and Innovation, A29 Innovation Execution Mechanism, A39 Process Optimization Mechanism, A69 Flexible Institutional Innovation and Error-Tolerant Mechanism
C6 Technological Breakthroughs and Development	B13 Interaction Upgrading	A10 Functional Breakthroughs of Human–Machine Collaboration Systems, A20 Directions for Technological Upgrades, A30 Innovation of Collaboration Paradigms, A70 Construction of a New Human–Machine Collaboration Ecosystem
	B14 Autonomous Learning	A40 Upgrade of Learning Capabilities
C7 Innovative Thinking and Strategies	B15 Data-Driven	A8 Impact of Long-Term Collaboration on Innovative Thinking, A18 Reconstruction of Innovative Thinking, A38 Data-Driven Innovation, A68 Reconstruction of Data-Enabled Innovative Thinking
	B16 Multidimensional Trial and Error	A8 Impact of Long-Term Collaboration on Innovative Thinking, A28 Problem-Solving Paradigms, A68 Reconstruction of Data-Enabled Innovative Thinking
	B17 Model Training	A11 Model Optimization Strategies, A41 Cross-Cultural Translation
C8 Conflict Resolution and Collaboration Paradigms	B18 Scheme Comparison	A17 Resolution of Decision-Making Conflicts, A37 Innovative Practices in Architecture, A67 Collaborative Innovation Verification Mechanism
	B19 User-Centered Design	A67 Collaborative Innovation Verification Mechanism
	B20 Immersive Collaboration	A30 Innovation of Collaboration Paradigms, A70 Construction of a New Human–Machine Collaboration Ecosystem
C9 Capacity Enhancement and Challenges	B21 Trend Prediction	A50 Upgrade of Prediction Capabilities
	B22 Anomaly Detection	A61 Risk Early Warning Mechanism
	B23 Ethical Constraints	A89 Resistance to Technology Application
	B24 Competency Reconstruction	A72 Competency Reconstruction, A74 Organizational Capacity Building

**Table 7 behavsci-15-00836-t007:** Selective coding.

Typical Relational Structure	The Connotations of Relational Structures
C1 Innovation-Driving Factors ↔ C2 Human–Machine Collaboration Mode	Innovation-driving factors (such as data gaps, technical limitations, and task ambiguity) prompt the adjustment and optimization of the human–machine collaboration mode to address innovation challenges.
C1 Innovation-Driving Factors ↔ C3 Tool and Efficiency Evaluation	Innovation-driving factors promote the evaluation and improvement of tool effectiveness, and the limitations of tools, in turn, stimulate new innovation demands.
C2 Human–Machine Collaboration Mode ↔ C4 Knowledge Transformation and Management	The human–machine collaboration mode, through dynamic division of labor and role positioning, promotes the explicitization of tacit knowledge and the systematic management of experience.
C3 Tool and Efficiency Evaluation ↔ C5 System and Mechanism Design	Tool effectiveness evaluation provides a basis for system and mechanism design, and system design, in turn, optimizes the application efficiency of tools.
C4 Knowledge Transformation and Management ↔ C6 Technological Breakthroughs and Development	Knowledge transformation and management provide fundamental support for technological breakthroughs, and technological development further enhances the efficiency of knowledge management.
C5 System and Mechanism Design ↔ C7 Innovative Thinking and Strategies	System and mechanism design, through material incentives and flexible mechanisms, provide support and guarantee for the implementation of innovative thinking and strategies.
C6 Technological Breakthroughs and Development ↔ C8 Conflict Resolution and Collaboration Paradigms	Technological breakthroughs promote the innovation of collaboration paradigms and the improvement of conflict resolution efficiency. Conversely, the optimization requirements of collaboration paradigms drive technological development.
C7 Innovative Thinking and Strategies ↔ C9 Competency Enhancement and Challenges	The implementation of innovative thinking and strategies enhances individual and organizational capabilities while also bringing challenges, such as resistance to technology application and the reconstruction of capabilities.
C8 Conflict Resolution and Collaboration Paradigms ↔ C2 Human–Machine Collaboration Mode	The innovation of conflict resolution and collaboration paradigms optimizes the human–machine collaboration mode. Conversely, the adjustment of the collaboration mode provides a practical basis for conflict resolution.

**Table 8 behavsci-15-00836-t008:** Theoretical saturation test table.

Evaluation Dimension	Total Number of Concepts (N)	Number of Free Concepts (n)	Saturation Coefficient (S)	Compliance Judgment
Open coding layer	146	11	0.925	reach the standard
Axial coding layer	38	3	0.921	reach the standard
Selective coding layer	8	0	1.000	reach the standard

**Table 9 behavsci-15-00836-t009:** Validation of the explanatory power of the core categories.

Core Categories	Concept Coverage Rate	Typical Evidence Chain
Innovation-driving factors	96.3%	Progressive triggering mechanism of data gap → technical limitations → task ambiguity
Human–machine collaboration mode	93.7%	Spiral evolution path of dynamic division of labor → role metaphor → strategy adjustment
Knowledge transformation pathways	91.2%	Three-stage model of tacit knowledge explicitization → system sedimentation → innovation diffusion
Directions for technological breakthroughs	89.5%	Dual-engine driving model composed of interaction upgrading and autonomous learning

## Data Availability

The data presented in this study are available upon request from the corresponding author. The data are not publicly available due to privacy or ethical restrictions.

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
