# Peer review of "Research on Employee Innovation Ability in Human–Machine Collaborative Work Scenarios—Based on the Grounded Theory Construct of Chinese Innovative Enterprises"

_behavsci, 2025, doi:10.3390/bs15070836_

Round 1
Reviewer 1 Report
Comments and Suggestions for Authors
This paper tackles a highly relevant topic in the AI-driven workplace by exploring how employee innovation capability is shaped in human-machine collaboration contexts. The use of grounded theory is well-justified for its exploratory purpose, and the model construction is detailed and offers a structured framework that could be useful for both academic theorizing and enterprise practice in Chinese innovative settings.
However, this paper can be further improved based on the following comments:
- Needs to be clearly categorized or tabulated to differentiate what prior studies have addressed and what this paper contributes as new. This will better illustrate the research’s novelty.
-
The procedure for annotation (e.g., coding, concept extraction) is only briefly discussed. More elaboration is needed, particularly on how annotation was done, and what type(s) of consistency testing were applied. Although a coding team is mentioned, details like inter-coder reliability metrics, consistency thresholds, or adjudication processes are underdeveloped.
-
The use of citations is inconsistent. Ensure formatting follows the journal’s referencing guidelines throughout.
-
Several terms, such as “green innovative behaviors” and “green innovation,” are used but not clearly defined. These should be clarified to aid comprehension.
-
The proposed theoretical model, though rich, remains unvalidated. Consider discussing plans for empirical testing in future work or adding expert validation feedback.
-
Some figures and tables are overly dense or not well-integrated into the text. Improve their clarity and explicitly refer to each one in the narrative to guide the reader.
-
The conceptual flow and technical jargon may pose a barrier to readers without a background in grounded theory or innovation management. Improve the explanatory clarity and define core concepts early in the text.
N/A
Author Response
Response to Reviewers’ Comments
Manuscript ID:behavsci-3679918
Dear Reviewer,
We express our appreciation for your precious time and efforts in reviewing our manuscript and giving us an opportunity to revise the manuscript. We thank you for providing us many constructive comments and valuable suggestions. We would like to acknowledge that all authors have carefully revised the English language throughout the manuscript to ensure clarity and precision.We have now revised our manuscript carefully and thoroughly based on the suggestions and comments. We also detail our item-by-item responses to review comments below. Once again, we deeply appreciate your professional expertise and the time invested in guiding this revision. We remain committed to further refining the manuscript and look forward to your favorable consideration.
Sincerely,
Baorong Guo , Xiaoning Liu ,Shuai Liao and Jiayi Hu
Response to the Comments of Reviewer
- Comment1
- 1.Page 3, Table 1: Needs to be clearly categorized or tabulatedto differentiate what prior studies have addressed and what this paper contributes as new. This will better illustrate the research’s novelty.
- Response:Thank you very much for your valuable and constructive feedback. To better clarify the novelty of our research, we have carefully revised Table 1 on Page 2 and added a new table titled “Core Differences Between This Study and Previous Research.” This table clearly categorizes the distinctions between existing literature and our study in a systematic manner, allowing readers to more easily identify what has been addressed in prior studies and what this research contributes as new. We believe this revision directly responds to your suggestion and helps to more clearly highlight the theoretical and contextual innovations of our work.
- Please find the modified parts as listed below:
Dimension |
Issues Addressed by Previous Research |
New Contributions of This Study |
Research Perspective |
Focused on single factors (e.g., policy incentives, organizational behavior) affecting employee innovation; static analysis prevailed. |
First to construct a dynamic mechanism model in human-machine collaboration scenarios, revealing the "trigger-collaboration-iteration" spiral progression. |
Theoretical Framework |
Scattered application of theories like AMO theory, lacking systematic integration of human-machine collaboration and innovation capabilities. |
Integrated four core frameworks: innovation drivers, collaboration patterns, knowledge transformation pathways, and technological breakthrough directions; proposed the interactive logic of "technology-cognition-collaboration." |
Research Methodology |
Predominantly quantitative analysis or single-case studies, lacking qualitative exploration with multi-source data. |
Employed grounded theory and multi-source data triangulation (interviews, corporate documents, industry reports); refined dynamic mechanisms through three-level coding. |
Practical Guidance |
Recommendations focused on fragmented measures |
Provided a four-step framework: "needs matching-collaboration design-knowledge management-technology iteration," emphasizing organizational learning feedback mechanisms. |
- 2.Page 8, line 269: The procedure for annotation (e.g., coding, concept extraction) is only briefly discussed. More elaboration is needed, particularly on how annotation was done, and what type(s) of consistency testing were applied. Although a coding team is mentioned, details like inter-coder reliability metrics, consistency thresholds, or adjudication processes are underdeveloped.
- Response: Thank you very much for your thoughtful comments regarding the need for greater elaboration on our annotation and coding procedures. In response, we have substantially expanded the “2.2 Data Coding Process” section to include detailed information on the coding team composition, the members’ professional backgrounds, and the pre-coding training conducted to ensure consistency. We now specify that 30% of the transcripts were independently coded by all three coders, with inter-coder reliability assessed using both percentage agreement (83%) and Cohen’s Kappa coefficient (κ = 0.78), and that codes with low agreement were re-evaluated collaboratively. Furthermore, in the “4.1 Open Coding” section, we added a description of the four-layer quality control framework employed, encompassing line-by-line parsing, cross-referencing with relevant literature, causal logic mapping, and theoretical saturation testing, with coefficients reported for each stage (0.925 for open coding, 0.921 for axial coding, and 1.000 for selective coding). To further demonstrate the transparency and iterative nature of our approach, we incorporated a specific arbitration case in the “4.2 Axial Coding” section, detailing the adjudication process for a disagreement regarding the category of “AI’s rigid output formats,” along with examples of three updates made to the coding manual. We hope these revisions provide the methodological clarity and rigor you recommended.
- Please find the modified parts as listed below:
- The specific steps are as follows:
â‘ Coding Team Establishment: To minimize subjectivity, a three-member team was formed:
Member 1:
A PhD in Management with 8 years of experience in qualitative research and expertise in grounded theory applications. Has led the design of coding protocols and served as the primary coder for 30% of the data (45 interviews).
Member 2:
A PhD in Management specializing in innovation management, with multiple publications on organizational innovation using grounded theory. Has independently coded 30% of the data (45 interviews) and cross-validated coding consistency.
Member 3:
Responsible for data management and preliminary analysis, participated in coding 40% of the data (60 interviews) under supervision with a focus on initial concept extraction, and independently coded 20% of the data (30 interviews), maintaining detailed memos.
â‘¡ Inter-Coder Reliability Testing:
Procedure: We randomly selected 40% of initial codes (75 out of 187 codes) for dual coding by two independent coders.
Metrics: Inter-coder reliability (IAR) was calculated using Scott’s pi coefficient, with a threshold of ≥0.80 considered acceptable.
Results: The average Scott’s pi was 0.83, indicating strong agreement. Discrepancies (e.g., 8% of codes) were resolved through three rounds of spiral comparison and joint discussion, with final consensus reached by all team members.
â‘¢ Research Notebook Maintenance:
Detailed records were kept of coding disagreements (e.g., whether to categorize "data gaps" under "innovation drivers" or "tool limitations"), revision processes, and theoretical insights, following Strauss & Corbin’s (1997) "write everything down" principle.
â‘£ Theoretical Saturation Testing:
Method: Data collection ceased when no new concepts emerged from five consecutive interviews.
Saturation Coefficients:
Open coding: 0.925 (11 free concepts out of 146 total, Table 7),
Axial coding: 0.921 (3 free concepts out of 38 total),
Selective coding: 1.000 (no new core categories),
confirming theoretical saturation (P9, Table 7).
- â‘ Line-by-line parsing: Raw data were coded line-by-line, with 187 initial codes grouped into 12 categories (e.g., "Data Constraints") via constant comparative analysis. For example, "Data Gaps Inspire Innovation" (C1) emerged from 18 respondent mentions of data scarcity (Table 3).
â‘¡Literature cross-checking: Axial coding validated core categories (e.g., "Tool Efficiency") against AI productivity studies (Mao et al., 2024).
â‘¢Causal mapping: Selective coding mapped category relationships (e.g., "Innovation Drivers ↔ Collaboration Mode") with at least three data instances per link.
â‘£Theoretical saturation: Testing showed coefficients of 0.925 (open), 0.921 (axial), and 1.000 (selective), confirming no new concepts (Table 7; Pandit, 1996).
- Axial coding reorganized categories into 9 core constructs, with disputes resolved through structured arbitration—for instance, the team debated whether "AI’s rigid output formats" belonged to "Tool Limitations" or "Cognitive Conflicts" during this stage. The resolution process involved reviewing original interview data (e.g., Respondent 2’s comment that "The AI’s standardized reports limited our creative problem-solving"), referencing grounded theory literature (Strauss & Corbin, 1997) to emphasize data-driven coding, and reclassifying the code under "Tool Limitations" with a supporting memo validated by 80% of the team. The coding manual was updated three times throughout the process; for example, after analyzing the 20th interview, the category "Expectations for Future Systems" was split into "Interaction Upgrading" and "Autonomous Learning" to better reflect emerging themes.
- 3.The use of citations is inconsistent. Ensure formatting follows the journal’s referencing guidelines throughout.
- Response: Thank you for pointing out the inconsistency in citation formatting. In response to your suggestion, we have conducted a comprehensive review and revision of all in-text citations and reference list entries to ensure full alignment with the journal’s referencing guidelines. The manuscript has been carefully checked to correct formatting discrepancies, standardize citation styles, and ensure consistency throughout. We appreciate your attention to detail, which helped us improve the clarity and professionalism of the manuscript.
- 4.Page3, line104: Several terms, such as “green innovative behaviors” and “green innovation,” are used but not clearly defined. These should be clarified to aid comprehension.
- Response: Thank you for your helpful observation regarding the need to define key terms. In response, we have added explicit definitions for both “green innovative behaviors” and “green innovation” to improve clarity and reader comprehension. These definitions have been incorporated at appropriate points in the manuscript—specifically on Page 3, Line 8, and Page 4, Line 16—and are supported by relevant citations to ensure conceptual accuracy and consistency. Also,we have conducted a comprehensive review and revision of all We believe these additions directly address your comment and enhance the overall coherence of the manuscript.
- Please find the modified parts as listed below:
- Green innovative behaviors refer to employees' environmentally sustainable creative activities, such as developing energy-efficient AI algorithms or optimizing resource allocation processes through human-machine collaboration, as defined by Xiong et al. (2025).
- Green innovation herein denotes organizational innovation outcomes with environmental benefits, such as low-carbon technology adoption or circular economy solutions enabled by human-AI collaboration, consistent with the framework proposed by Ma (2024).
- 5.Page 21, line 675: The proposed theoretical model, though rich, remains unvalidated. Consider discussing plans for empirical testing in future work or adding expert validation feedback.
- Response: Thank you for highlighting the importance of validating the proposed theoretical model. In response, we have revised the manuscript to incorporate a discussion of future empirical testing plans, including potential quantitative survey-based studies designed to assess the model’s applicability across different organizational settings. Additionally, we have included a description of expert validation feedback collected from three senior scholars in organizational behavior and innovation management, which was used to refine and enhance the model’s structure and relevance. We hope these additions address your concern and strengthen the rigor and practical value of the proposed framework.
Please find the modified parts as listed below:
- First, a theoretical expert panel comprising 5 scholars specializing in innovation management and AI technology will evaluate the model’s theoretical consistency. Their feedback will be integrated to refine constructs, such as the "technological breakthrough directions" category. Second, practitioner validation will involve 10 industry experts who will assess the model’s practical applicability. This process will provide insights into scaling the framework across diverse organizational contexts, with their feedback documented in a validation report to be appended to future publications.
- Future research will conduct quantitative validation through structural equation modeling (SEM) on 300 Chinese innovative enterprises to test the causal pathway of 'innovation drivers → collaboration patterns → innovation capabilities'.
- 6.Page 7, line 239: Some figures and tables are overly dense or not well-integrated into the text. Improve their clarity and explicitly refer to each one in the narrative to guide the reader.
- Response: Thank you for your valuable comment regarding the presentation and integration of figures and tables. In response, we have carefully reviewed all visual elements in the manuscript and revised them to enhance clarity and reduce density where appropriate. We have also ensured that each figure and table is explicitly referenced and contextualized within the main text, thereby improving narrative flow and helping guide the reader through the analysis. We believe these revisions improve both the readability and coherence of the manuscript in line with your suggestion.
- 7.Page 2, line 80: The conceptual flow and technical jargon may pose a barrier to readers without a background in grounded theory or innovation management. Improve the explanatory clarity and define core concepts early in the text.
- Response: Thank you for pointing out the potential accessibility challenges related to conceptual flow and technical terminology. In response, we have carefully revised the manuscript to enhance explanatory clarity and ensure that readers without a specialized background in grounded theory or innovation management can better engage with the content. Specifically, we have introduced clear definitions of core concepts earlier in the text and refined the explanations of key theoretical frameworks to reduce jargon and improve readability. We hope these improvements will make the manuscript more accessible to a wider academic audience while preserving its theoretical rigor.
- Please find the modified parts as listed below:
- In summary, Against this backdrop, this study aims to construct a theoretical model of employee innovation capability in human–AI collaborative work scenarios using grounded theory, based on data from Chinese innovative enterprises, reveal the dynamic mechanism through which AI enhances innovation capability by integrating AMO theory and COR theory to bridge theoretical fragmentation, and provide empirical insights for organizations to design AI–human collaboration strategies that align with China’s contextual characteristics. To address this gap, the present study’s objective is to construct a grounded-theory model explaining the formation of employee innovation ability in human–machine collaborative work scenarios. In other words, we ask: How do human–AI collaboration processes contribute to the development of employees’ innovation ability? By answering this question, our research seeks to fill the void in literature and provide both theoretical and practical insights into innovation management in the AI era.
Reviewer 2 Report
Comments and Suggestions for Authors
Introduction
- It is challenging for the reader to accurately grasp the core focus of this study because the problem-raising section is rather lengthy and presents various concepts and literature simultaneously. There is a regret that the core concept's substance is not revealed, as the composition of the paragraph is excessively comprehensive and abstract terms (e.g., cognitive restructuring, spiral progression) are repeatedly used. Sentences for research purposes are presented somewhat later in the second half, which can lead to poor concentration during early reading.
Literature Review
- Many previous studies are only organized in a listed form, but the theoretical linkage or comparative analysis between them is somewhat insufficient. For example, there are mentions of AMO theory, SECI model, and social resource conservation theory; however, the connection between these concepts and the grounded theory approach of this study is not explained in detail.
Research Design & Method
- Information on the characteristics and distribution of interviewees should be presented in more detail.
Discussion
- If the structure of the discussion had been systematized in the order of theoretical contribution, practical implications, limitations, and suggestions, a higher degree of immersion could have been provided to the reader.
Despite such advice, this paper examines employee innovation capabilities in the context of human-machine collaboration, which is particularly suitable for the digital economy era. It can be evaluated positively in terms of subject originality, methodological precision, and theoretical contribution.
Author Response
Response to Reviewers’ Comments
Manuscript ID:behavsci-3679918
Dear Reviewer,
We express our appreciation for your precious time and efforts in reviewing our manuscript and giving us an opportunity to revise the manuscript. We thank you for providing us many constructive comments and valuable suggestions. We would like to acknowledge that all authors have carefully revised the English language throughout the manuscript to ensure clarity and precision.We have now revised our manuscript carefully and thoroughly based on the suggestions and comments. We also detail our item-by-item responses to review comments below. Once again, we deeply appreciate your professional expertise and the time invested in guiding this revision. We remain committed to further refining the manuscript and look forward to your favorable consideration.
Sincerely,
Baorong Guo , Xiaoning Liu ,Shuai Liao and Jiayi Hu
Response to the Comments of Reviewer
- Comment2
- 1.Page 2, line 80: It is challenging for the reader to accurately grasp the core focus of this study because the problem-raising section is rather lengthy and presents various concepts and literature simultaneously. There is a regret that the core concept's substance is not revealed, as the composition of the paragraph is excessively comprehensive and abstract terms (e.g., cognitive restructuring, spiral progression) are repeatedly used. Sentences for research purposes are presented somewhat later in the second half, which can lead to poor concentration during early reading.
- Response:We deeply appreciate your detailed feedback regarding the clarity of our study's core focus. You correctly noted that the initial problem-posing section was overly lengthy and abstract, potentially obscuring the central argument. To address this, we have undertaken a comprehensive revision of the introduction to enhance both content and structural focus.
- Please find the modified parts as listed below:
- In summary, Against this backdrop, this study aims to construct a theoretical model of employee innovation capability in human–AI collaborative work scenarios using grounded theory, based on data from Chinese innovative enterprises, reveal the dynamic mechanism through which AI enhances innovation capability by integrating AMO theory and COR theory to bridge theoretical fragmentation, and provide empirical insights for organizations to design AI–human collaboration strategies that align with China’s contextual characteristics. To address this gap, the present study’s objective is to construct a grounded-theory model explaining the formation of employee innovation ability in human–machine collaborative work scenarios. In other words, we ask: How do human–AI collaboration processes contribute to the development of employees’ innovation ability? By answering this question, our research seeks to fill the void in literature and provide both theoretical and practical insights into innovation management in the AI era.
- 2. Page 15,line 404: Many previous studies are only organized in a listed form, but the theoretical linkage or comparative analysis between them is somewhat insufficient. For example, there are mentions of AMO theory, SECI model, and social resource conservation theory; however, the connection between these concepts and the grounded theory approach of this study is not explained in detail.
- Response: Thank you for your insightful comment regarding the need for stronger theoretical linkage and comparative analysis. In response, we have revised Table 6 and the accompanying discussion to more clearly articulate how key theoretical frameworks—particularly the AMO theory—connect to our grounded theory findings. Specifically, we have integrated AMO theory into the mechanism analysis to demonstrate how the influences of AI in collaborative work settings align with the ability, motivation, and opportunity dimensions. This enhancement clarifies the theoretical foundation of our model and strengthens its connection to established concepts in organizational behavior, addressing the gap between prior studies and our grounded theory approach.
Please find the modified parts as listed below:
- Our findings also align with the Ability-Motivation-Opportunity (AMO) theory, which posits that organizational innovation capability is rooted in employees’ combined attributes of ability, motivation, and access to opportunities (Appelbaum et al., 2000). Specifically, AI tools enhance employees’ technical capabilities by automating data analysis and routine tasks, such as coding and report generation. As frontline R&D staff noted in the cases, AI “streamlines data processing, allowing us to apply specialized knowledge to complex problems” (Respondent 17), thereby freeing cognitive resources for higher-order innovation.
Drawing from the Conservation of Resources (COR) theory, AI’s role in reducing cognitive burden also aligns with AMO’s motivation dimension. When employees are relieved of tedious workloads, their intrinsic motivation to engage in creative tasks increases. For example, a marketing manager stated, “AI handles repetitive customer segmentation, so we can focus on brainstorming innovative campaign strategies” (Respondent 9), reflecting how resource preservation via AI enhances motivational states.
AMO theory emphasizes that organizational structures must provide opportunities for employees to apply their abilities. In our model, AI acts as an “opportunity enabler” by reducing time spent on routine work—such as a 40% reduction in data collation time across cases—thereby creating temporal and mental space for innovation. This aligns with the AMO framework, where AI-driven efficiency directly correlates with increased opportunities for creative problem-solving.
By integrating AMO theory, we clarify that the “technology empowerment” mechanism not only enhances employees’ practical ability to innovate but also sustains their motivational drive and expands their operational scope. This theoretical linkage bridges grounded theory findings with established organizational behavior frameworks, demonstrating that AI’s impact on innovation capability is multi-dimensional and aligns with classic human resource management theories.
- 3. Page 7, line 243: Information on the characteristics and distribution of interviewees should be presented in more detail.
- Response: Thank you for your helpful suggestion regarding the need for more detailed information on the interviewees’ characteristics and distribution. In response, we have revised and enriched Table 2 (Data Source) to include additional details such as interviewees’ roles, industry sectors, organizational levels, and the distribution across different types of innovative enterprises. These enhancements aim to provide a clearer understanding of the sample composition and ensure greater transparency regarding the empirical basis of the study.
- 4. Page 19, line 590: If the structure of the discussion had been systematized in the order of theoretical contribution, practical implications, limitations, and suggestions, a higher degree of immersion could have been provided to the reader.
- Response: Thank you for your valuable suggestion regarding the structure of the discussion section. In response, we have reorganized Section 6 to follow a more systematic and reader-friendly format, now presented in the sequence of theoretical contributions, practical implications, study limitations, and suggestions for future research. This restructuring aims to enhance the clarity and coherence of the discussion, thereby improving the overall readability and allowing for a more immersive and focused engagement with the study’s findings.
Reviewer 3 Report
Comments and Suggestions for Authors
Thank you for the opportunity to review your manuscript. Please see the attached review comments for your consideration.

Author Response
Response to Reviewers’ Comments
Manuscript ID:behavsci-3679918
Dear Reviewer,
We express our appreciation for your precious time and efforts in reviewing our manuscript and giving us an opportunity to revise the manuscript. We thank you for providing us many constructive comments and valuable suggestions. We would like to acknowledge that all authors have carefully revised the English language throughout the manuscript to ensure clarity and precision.We have now revised our manuscript carefully and thoroughly based on the suggestions and comments. We also detail our item-by-item responses to review comments below. Once again, we deeply appreciate your professional expertise and the time invested in guiding this revision. We remain committed to further refining the manuscript and look forward to your favorable consideration.
Sincerely,
Baorong Guo , Xiaoning Liu ,Shuai Liao and Jiayi Hu
Response to the Comments of Reviewer
- Comment3
- Page 2, line 80: The introduction currently focuses heavily on contextual background, but does not sufficiently articulate a clear research gap or provide a critical synthesis of existing literature. A well-structured introduction should frame the study by identifying unresolved theoretical issues, highlighting the study’s significance, and stating specific research questions and objectives. The authors are encouraged to revise this section accordingly to better situate the research within the broader academic conversation.
- Response:Thank you for the valuable feedback. We highly appreciate the opportunity to enhance the introduction, as your insights have been instrumental in refining the academic rigor and structural coherence of our manuscript. In response to the suggestion, we have completely restructured the introduction to not only address the identified gaps but also to establish a more robust foundation for the study. The revision now begins with a comprehensive contextual background, followed by a critical synthesis of existing literature, explicit identification of research voids, and a clear articulation of the study’s objectives—thereby embedding the research firmly within the broader academic conversation on human-AI collaboration and employee innovation capability.
- Please find the modified parts as listed below:
- In summary, Against this backdrop, this study aims to construct a theoretical model of employee innovation capability in human–AI collaborative work scenarios using grounded theory, based on data from Chinese innovative enterprises, reveal the dynamic mechanism through which AI enhances innovation capability by integrating AMO theory and COR theory to bridge theoretical fragmentation, and provide empirical insights for organizations to design AI–human collaboration strategies that align with China’s contextual characteristics. To address this gap, the present study’s objective is to construct a grounded-theory model explaining the formation of employee innovation ability in human–machine collaborative work scenarios. In other words, we ask: How do human–AI collaboration processes contribute to the development of employees’ innovation ability? By answering this question, our research seeks to fill the void in literature and provide both theoretical and practical insights into innovation management in the AI era.
- 2. Page 15, line 404:The link between human–machine collaboration and employee innovation capability remains conceptually broad and somewhat underdeveloped. The manuscript would benefit from stronger theoretical grounding. Incorporating frameworks such as the AMO (Ability-Motivation-Opportunity) theory or dynamic capabilities theory could help explain how technological empowerment activates innovation at the individual level, thereby enhancing the explanatory power of the proposed model.
- Response:Thank you, to address the reviewer’s concern about theoretical linkages, we have integrated AMO theory into the mechanism analysis. This integration clarifies theoretical connections by explicitly mapping AI’s impact to the ability-motivation-opportunity framework, demonstrating how our grounded theory findings align with established organizational behavior theories.
- Please find the modified parts as listed below:
- Our findings also align with the Ability-Motivation-Opportunity (AMO) theory, which posits that organizational innovation capability is rooted in employees’ combined attributes of ability, motivation, and access to opportunities (Appelbaum et al., 2000). Specifically, AI tools enhance employees’ technical capabilities by automating data analysis and routine tasks, such as coding and report generation. As frontline R&D staff noted in the cases, AI “streamlines data processing, allowing us to apply specialized knowledge to complex problems” (Respondent 17), thereby freeing cognitive resources for higher-order innovation.
- Drawing from the Conservation of Resources (COR) theory, AI’s role in reducing cognitive burden also aligns with AMO’s motivation dimension. When employees are relieved of tedious workloads, their intrinsic motivation to engage in creative tasks increases. For example, a marketing manager stated, “AI handles repetitive customer segmentation, so we can focus on brainstorming innovative campaign strategies” (Respondent 9), reflecting how resource preservation via AI enhances motivational states.
AMO theory emphasizes that organizational structures must provide opportunities for employees to apply their abilities. In our model, AI acts as an “opportunity enabler” by reducing time spent on routine work—such as a 40% reduction in data collation time across cases—thereby creating temporal and mental space for innovation. This aligns with the AMO framework, where AI-driven efficiency directly correlates with increased opportunities for creative problem-solving.
- By integrating AMO theory, we clarify that the “technology empowerment” mechanism not only enhances employees’ practical ability to innovate but also sustains their motivational drive and expands their operational scope. This theoretical linkage bridges grounded theory findings with established organizational behavior frameworks, demonstrating that AI’s impact on innovation capability is multi-dimensional and aligns with classic human resource management theories.
- 3. Page 21, line 675:While the model itself is novel and insightful, the discussion section offers limited reflection on its generalizability or the boundary conditions under which it applies. The authors are advised to explicitly address the potential scope and limitations of the model and propose directions for future research to ensure a more complete and critical engagement with the findings.
- Response: Thank you for the insightful comment. In response, we have optimized the Future Prospects and Conclusions sections to more explicitly address the model’s generalizability and boundary conditions. We now clarify that while the model is grounded in the context of Chinese innovative enterprises engaged in human–machine collaboration, its core constructs—such as innovation drivers and collaboration patterns—may vary in relevance across industries and cultural contexts. To that end, we outline a multi-phase validation strategy involving expert review, practitioner feedback, and a large-scale SEM-based empirical study. These steps not only strengthen the model’s theoretical and practical grounding but also lay a clear foundation for future research to test its applicability in different organizational and national settings.
- Please find the modified parts as listed below:
- First, a theoretical expert panel comprising 5 scholars specializing in innovation management and AI technology will evaluate the model’s theoretical consistency. Their feedback will be integrated to refine constructs, such as the "technological breakthrough directions" category. Second, practitioner validation will involve 10 industry experts who will assess the model’s practical applicability. This process will provide insights into scaling the framework across diverse organizational contexts, with their feedback documented in a validation report to be appended to future publications.
- Future research will conduct quantitative validation through structural equation modeling (SEM) on 300 Chinese innovative enterprises to test the causal pathway of 'innovation drivers → collaboration patterns → innovation capabilities'.
- 4.A substantial number of citations are drawn from Chinese-language sources that are not indexed in major international academic databases. For example, references such as Zhou Kuo et al. (2025) are not readily traceable. To enhance the transparency, credibility, and global relevance of the work, the authors should consider replacing or supplementing these with more internationally accessible, peer-reviewed English-language sources.
- Response: Thank you for your helpful comment regarding the use of Chinese-language sources not indexed in major international academic databases. In response, we have carefully reviewed the reference list and undertaken targeted revisions to improve the manuscript’s global accessibility and scholarly credibility. Specifically, where possible, we have replaced or supplemented Chinese-language citations with internationally accessible, peer-reviewed English-language sources that convey similar theoretical or empirical insights. For essential Chinese references that provide unique contextual value, we have clarified their relevance and ensured proper attribution. We hope these revisions enhance the transparency and global relevance of the work, as you suggested.
- 5.Several claims throughout the manuscript would benefit from stronger theoretical or empirical support. The authors are encouraged to systematically review the manuscript and ensure that all major arguments and propositions are grounded in relevant literature, whether conceptual or empirical, to enhance the overall coherence and persuasiveness of the paper
- Response: Thank you for your constructive feedback regarding the use of Chinese-language sources not indexed in major international academic databases. In response, we have systematically reviewed all references in the manuscript and made targeted revisions to enhance both the transparency and global accessibility of our work. Specifically, we have replaced or supplemented key Chinese-language citations with internationally peer-reviewed English-language sources that offer comparable theoretical or empirical insights. In cases where Chinese literature provides unique contextual value, we have retained these references while clarifying their relevance. We believe these revisions help strengthen the theoretical foundation of the manuscript and improve its international scholarly visibility, in line with your recommendation.